# Integrating hypertension and HIV care in Namibia: A quality improvement collaborative approach

Apollo Basenero[1], Julie Neidel[2], Daniel J. Ikeda[3], Hilaria Ashivudhi[4], Simbarashe Mpariwa[1], Jacques W. N. Kamangu[1], Mireille A. Mpalang Kakubu[1], Linea Hans[4], Gram Mutandi[4], Suzanne Jed[5], Francina Tjituka[1], Ndapewa Hamunime[1], Bruce D. Agins[2]*

1 Ministry of Health and Social Services, Windhoek, Namibia, 2 Institute for Global Health Sciences, University of California, San Francisco, California, United States of America, 3 Harvard Medical School, Boston, Massachusetts, United States of America, 4 U.S. Centers for Disease Control and Prevention, Windhoek, Namibia, 5 Office of the U.S. Global AIDS Coordinator and Health Diplomacy, Pretoria, South Africa

* bruce.agins@ucsf.edu

**Data Availability Statement:** All relevant data are within the paper and its Supporting information files.

## Abstract

### Background

Hypertension (HTN) is highly prevalent among people with HIV (PWH) in Namibia, but screening and treatment for HTN are not routinely offered as part of HIV care delivery. We report the implementation of a quality improvement collaborative (QIC) to accelerate integration of HTN and HIV care within public-sector health facilities in Namibia.

### Methods

Twenty-four facilities participated in the QIC with the aim of increasing HTN screening and treatment among adult PWH (>15 years). HTN was defined according to national treatment guidelines (i.e., systolic blood pressure >140 and/or diastolic blood pressure >90 across three measurements and at least two occasions), and decisions regarding initiation of treatment were made by physicians only. Teams from participating hospitals used quality improvement methods, monthly measurement of performance indicators, and small-scale tests of change to implement contextually tailored interventions. Coaching of sites was performed on a monthly basis by clinical officers with expertise in QI and HIV, and sites were convened as part of learning sessions to facilitate diffusion of effective interventions.

### Results

Between March 2017 and March 2018, hypertension screening occurred as part of 183,043 (86%) clinical encounters at participating facilities. Among 1,759 PWH newly diagnosed with HTN, 992 (56%) were initiated on first-line treatment. Rates of treatment initiation were higher in facilities with an on-site physician (61%) compared to those without one (51%). During the QIC, facility teams identified fourteen interventions to improve HTN screening

**Funding:** The project described in this article was funded by the U.S. Department of Health and Human Services, Health Resources and Services Administration under cooperative agreement #U1NHA08599. The views expressed in this publication are solely the opinions of the authors and do not necessarily reflect the official policies of the U.S. Department of Health and Human Services, the Health Resources and Services Administration, nor does mention of the department or agency name imply endorsement by the U.S. Government.

**Competing interests:** The authors have declared that no competing interests exist.

and treatment. Among barriers to implementation, teams pointed to malfunctions of blood pressure machines and stock outs of antihypertensive medications as common challenges.

## Conclusions

Implementation of a QIC provided a structured approach for integrating HTN and HIV services across 24 high-volume facilities in Namibia. As rates of HTN treatment remained low despite ongoing facility-level changes, policy-level interventions—such as task sharing and supply chain strengthening—should be pursued to further improve delivery of HTN care among PWH beyond initial screening.

## Introduction

The scale up of antiretroviral therapy (ART) in sub-Saharan Africa (SSA) has led to steep declines in HIV-related mortality. In Namibia, a sparsely populated, middle-income country in SSA where an estimated 12% of the adult population is living with HIV, deaths from HIV have fallen by 22% since 2010 [1]. Although communicable diseases, including HIV, still comprise the leading causes of morbidity and mortality in Namibia [2], the prevalence of non-communicable diseases (NCD) such as cardiovascular disease, diabetes, and cancer has risen sharply, contributing to nearly 41% of all yearly deaths. Among preventable risk factors for NCDs, elevated blood pressure (i.e., hypertension) has been identified as the greatest contributor to NCD-related death and disability in Namibia [3]. Hypertension (HTN) affects an estimated one in five adults in Namibia [4, 5], yet remains underdiagnosed and undertreated, with only 47% reporting awareness of their condition and less than 20% achieving adequate control [5]. This gap has particular relevance to the care of PWH, who bear a disproportionate burden of HTN [6], and whose risk of cardiovascular disease exceeds that of the general population by a factor of two [7].

Over the past two decades, primary healthcare systems in Namibia and other countries in SSA with high burdens of HIV have evolved to support the engagement of people with HIV (PWH) in life-long care [8]. As decentralized platforms of longitudinal service delivery, these systems are uniquely suited to provide integrated care for the growing number of PWH with HTN [9]. Indeed, a cohort study in South Africa has observed an "ART advantage," showing that individuals engaged in HIV care were more likely to receive HTN screening and treatment than the general population [10]. Notably, integrated care has been previously implemented to address sexual and reproductive health in Namibia [11]. However, despite a clear opportunity for synergy and cost saving [12], implementation of integrated models of care for HIV and HTN in LMICs has been discouragingly slow. Several factors have been identified as barriers to implementation, including limited human resource capacity, fragile supply chains for diagnostic equipment and medication, weak health information systems, and ambiguous or nonexistent guidance on implementation [13]. In these settings, new interventions, models, and approaches to integrate HIV and HTN services at scale are needed, with specific attention to how context may influence their implementation and scalability [14–16].

Quality improvement collaboratives (QIC) have been promoted as an effective strategy for improving the quality of healthcare delivery across diverse contexts [17, 18]. Specifically, QICs seek to bridge the "know-do" gap—the gulf between clinical guidelines and their implementation—through capacity building, group problem-solving, shared leadership, and normative pressure [19]. In HIV service delivery [20], QICs have been implemented to address performance gaps in areas as varied as mother-to-child transmission [21, 22], provider-initiated

testing and counseling [23, 24], ART initiation [25], tuberculosis preventive therapy coverage [26], viral load monitoring [27], and stigma reduction [28]. However, no work to date has investigated how QICs may be applied to address gaps in scale up of integrated service delivery. In this work, we report the implementation of a QIC—the Namibia Project for Retention of Patients on ART (NAMPROPA)—whose objective was to improve uptake of HTN screening and treatment in routine HIV care in Namibia.

## Methods

### Implementation context

NAMPROPA was launched in November 2016 by Namibia's Ministry of Health and Social Services (MHSS) with the aim of improving care engagement, viral monitoring, viral suppression, and HTN screening and treatment in 24 ART facilities across three regions with high burdens of HIV—Khomas, Ohangwena, and Zambezi (Table 1). These sites were purposively selected based on their high relative patient volumes. All participating facilities are publicly funded and provide health services free-of-charge to over 50,000 PWH—nearly one third of all PWH receiving ART in Namibia's public sector. Although all facilities dispense ART on site, only a minority stock anti-HTN medications recommended as "first-line" by Namibia's Standard Treatment Guidelines. All facilities have a nurse on site, but only 33% have an on-site physician. In line with MHSS's approach to task sharing and decentralized HIV care delivery [29–31], facilities without an on-site physician remain in frequent contact with a regional medical officer. Importantly, nurses in Namibia can initiate ART in patients with a new HIV diagnosis, but only physicians can provide an initial prescription for anti-HTN therapy. Prior to the implementation of NAMPROPA, HTN screening and treatment were not routinely conducted as part of HIV service delivery in participating sites. Technical support for implementation of the QIC was provided by the University of California, San Francisco (UCSF), the Health Resources and Service Administration, and the U.S. Centers for Disease Control and Prevention.

### QIC approach

The design of NAMPROPA was adapted from the Institute for Healthcare Improvement's Breakthrough Series Model, an improvement science methodology in which participating sites

**Table 1. Site characteristics, by region.**

|  | Khomas | Ohangwena | Zambezi |
|---|---|---|---|
| Adult HIV prevalence (%) | 8.3% | 17.9% | 22.3% |
| Participating facilities, n | 7 | 10 | 7 |
| ART caseloads, n |  |  |  |
| 0–999 | 1 | 5 | 6 |
| 1000–4999 | 5 | 4 | 1 |
| 5000+ | 1 | 1 | 0 |
| Number of HCWs, n |  |  |  |
| 0–10 | 0 | 1 | 1 |
| 11–20 | 5 | 2 | 2 |
| >20 | 2 | 7 | 5 |
| On-site physician?, n |  |  |  |
| Yes | 6 | 3 | 2 |
| No | 1 | 7 | 5 |

ART–antiretroviral therapy; HCW–healthcare worker.

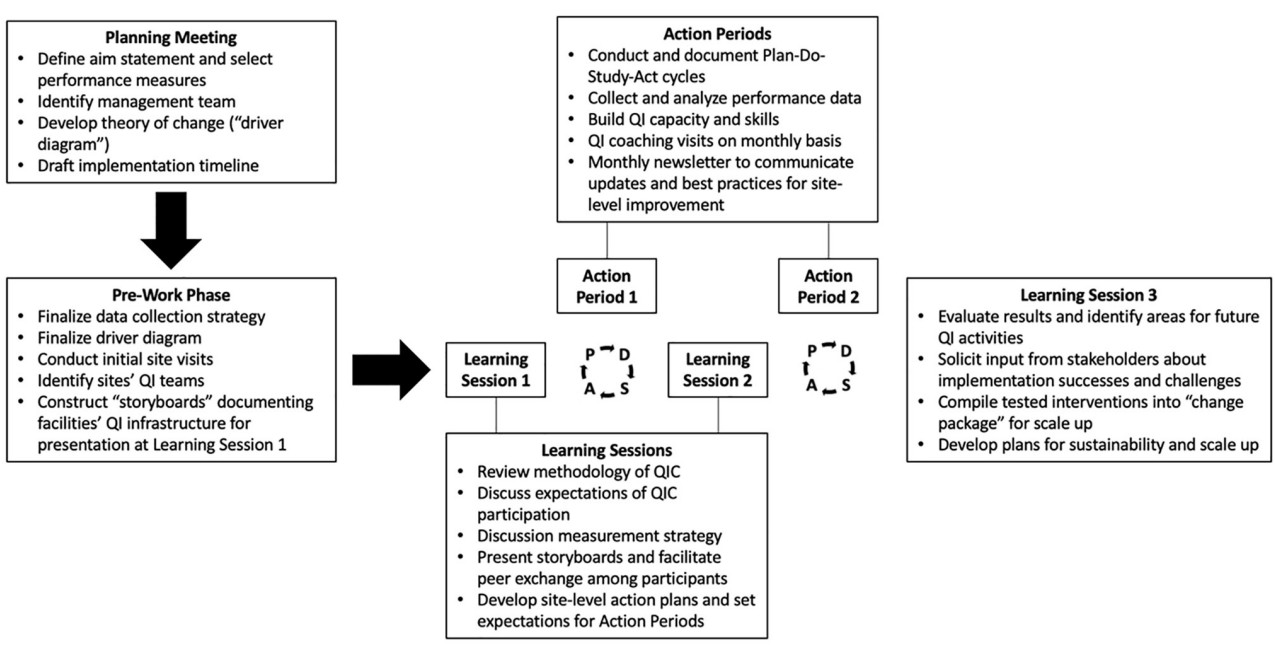

**Fig 1. NAMPROPA structure and activities.** QI–quality improvement; QIC–quality improvement collaborative.

are convened to apply quality improvement (QI) methods (e.g., Plan-Do-Study-Act cycles, cascade analysis, process mapping, root cause analysis) to the identification and improvement of gaps in a specific area of healthcare delivery [32]. QICs are complex, quasi-experimental interventions that collect repeated cross-sectional data to track implementation success [19]. Fig 1 presents specific QIC activities. In contrast to traditional QI approaches in which projects are implemented and evaluated at individual sites over a period of months, QICs feature more frequent performance measurement and smaller-scale tests of changes, enabling sites to test multiple interventions and share experiences with other participating facilities over an accelerated period of time. The accumulation of evidence-informed [33] interventions and performance improvements are significantly accelerated through QICs, leading to the compilation of a "change package" for use in scale-up activities. Since an overarching aim of NAMPROPA was to promote QI capacity-building, no facilities were assigned to serve as controls. MHSS had significant prior experience implementing QI programming [34–36], but no QIC had been implemented in Namibia prior to NAMPROPA.

## Coaching and support

Participating sites received coaching and support from regional and district clinical mentors, an existing cadre of physicians and nurses employed by MHSS with special expertise in HIV clinical care [29]. These mentors had been formally trained in QI principles, methods, and coaching through national MHSS-led QI seminars [34, 35]. As part of NAMPROPA activities, clinical mentors visited sites at a minimum of a monthly basis, integrating QI coaching into their routine activities to build capacity for HIV care and treatment with specific activities including review of sites' performance data, communication of NAMPROPA updates, and assistance with design, execution, and evaluation of Plan-Do-Study-Act cycles. Between coaching visits, sites convened QI meetings to plan and track implementation. Technical support for regional mentors in QI methods and coaching was provided by MHSS and UCSF with monthly check-in calls to monitor implementation and discuss challenges.

### Peer-to-peer exchange

Over 12 months, teams of three staff members from the 24 participating facilities, clinical mentors, and MHSS stakeholders were convened on three occasions as part of three-day, in-person learning sessions (LS). The objectives of LS were to gain proficiency in QI tools and methods, share examples of QI interventions, develop action plans, and prioritize future improvement activities. These LS also featured presentations from MHSS on current HTN treatment guidelines and protocols for referring patients with newly diagnosed HTN for treatment initiation. Between LS, ongoing peer-to-peer exchange was facilitated through participation in WhatsApp groups, email, and regional meetings.

### Data collection

Rates of HTN screening and treatment were reported by participating sites to MHSS on a monthly basis using a pre-programmed electronic workbook (Microsoft Excel 2013, Microsoft Corporation, Redmond, WA) equipped to produce automated run charts. These data were subsequently aggregated to reflect collaborative-wide and region-specific performance and presented to sites for benchmarking. Measures were defined in alignment with Namibia's Standard Treatment Guidelines for adults (>15 years) [37]. These guidelines define HTN as a blood pressure that consistently exceeds 140/90 mm Hg across three measurements and at least two occasions. In cases of mild HTN (<160/100 mm Hg), guidelines recommend provision of health education on mitigation of risk factors and monthly monitoring. Among patients with severe HTN ($\geq$160/100 mm Hg) or mild HTN that is resistant to initial intervention, monotherapy with a thiazide diuretic (i.e., amiloride/hydrochlorothiazide) is recommended as first-line treatment. To capture "change ideas" and site-reported implementation barriers, we conducted semi-structured interviews of 138 health workers (i.e., physicians, nurses, pharmacists, data clerks, health assistants, and field officers) as part of site visits and undertook a content analysis of these interviews and facility presentations as part of LS. The number of coaching visits and QI meetings were tracked by MHSS on a monthly basis. As part of the QIC's final learning session, participating facility teams compiled a "change package" of improvement interventions. During this process, participants from each region were convened to rank changes according to effectiveness (i.e., the magnitude of improvement on the basis of Plan-Do-Study-Act cycles) and perceived sustainability (i.e., how likely a change might be sustained in the long-term). Selected changes from each region were then presented to the large group, and a final list of changes was determined by group discussion and majority vote. Changes were subsequently collated according to domains of the Chronic Care Model—a heuristic for organizing changes that improve care for chronic conditions [38], including HIV [39].

### Ethical considerations

Implementation of NAMPROPA activities, including data collection and analysis, was approved by MHSS and the institutional review board of UCSF, receiving a determination of non-research. As no patient-level data were collected, the need for informed consent was waived. We used SQUIRE (Standards for QUality Improvement Reporting Excellence) guidelines to structure reporting of NAMPROPA implementation and results [40].

## Results

At baseline, no participating facilities were conducting screening for HTN. Prior to initiation of QIC activities, all sites were provided with blood pressure machines. Fig 2 reports the

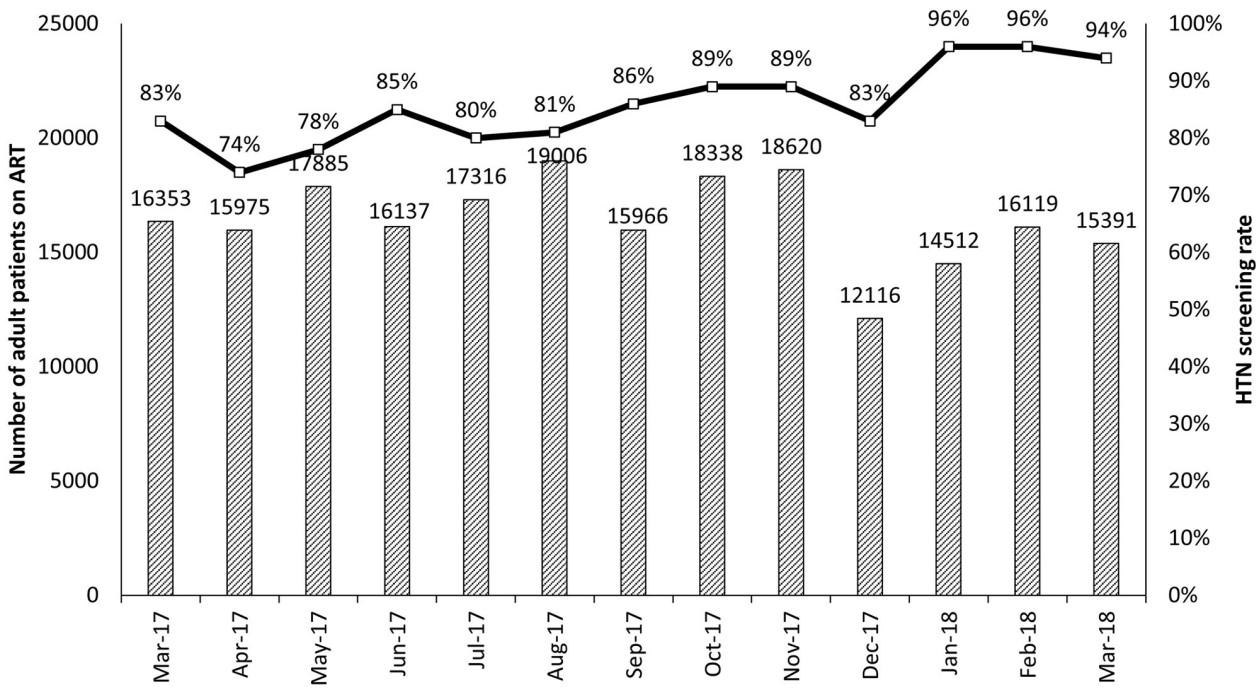

**Fig 2. HTN screening rate and number of adult patients on ART, March 2017–March 2018.** ART–antiretroviral therapy; HTN–hypertension.

number of monthly patient encounters during which HTN screening was provided. Of 213,714 clinical encounters at participating sites that occurred between March 2017 and March 2018, HTN screening was conducted as part of 183,043 (86%). Between LS 1 and 2, HTN screening occurred as a part of 80% of encounters, compared to 89% between LS 2 and 3. Barriers to HTN screening cited by facility teams included challenges recording and analyzing QIC measures; malfunctions of blood pressure machines; and difficulties following up with patients for multiple blood pressure readings.

Collection and reporting of data on HTN treatment did not begin until September 2017. Fig 3 displays the rate of treatment initiation among patients newly diagnosed with HTN each month. During the 7 months in which data on HTN treatment were collected, 1,759 patients on ART were newly diagnosed with HTN, and 992 (56%) were successfully initiated on treatment. Among sites with an on-site physician, treatment rates were 61%, compared to 51% among sites without an on-site physician. Several barriers to HTN treatment were noted by sites, including limited physician availability to initiate anti-HTN treatment; stock outs of medications; and difficulties tracking patient referrals to tertiary facilities for treatment initiation. Reassuringly, analyses of HIV-specific measures from NAMPROPA demonstrated no detrimental impact of HTN screening and treatment on care engagement, viral load monitoring or viral suppression [41].

During NAMPROPA, participating sites convened 204 QI meetings, and 100% received coaching visits at least monthly. As part of the QIC's third and final learning session, participating facility teams compiled a "change package" of improvement interventions that were assessed, on the basis of findings from PDSA cycles, to be both effective and feasible to sustainably implement. During NAMPROPA implementation, fourteen interventions were adopted to address HTN screening and treatment. Table 2 presents these interventions by Chronic Care Model domain. Overall, interventions in the domains of delivery system design and

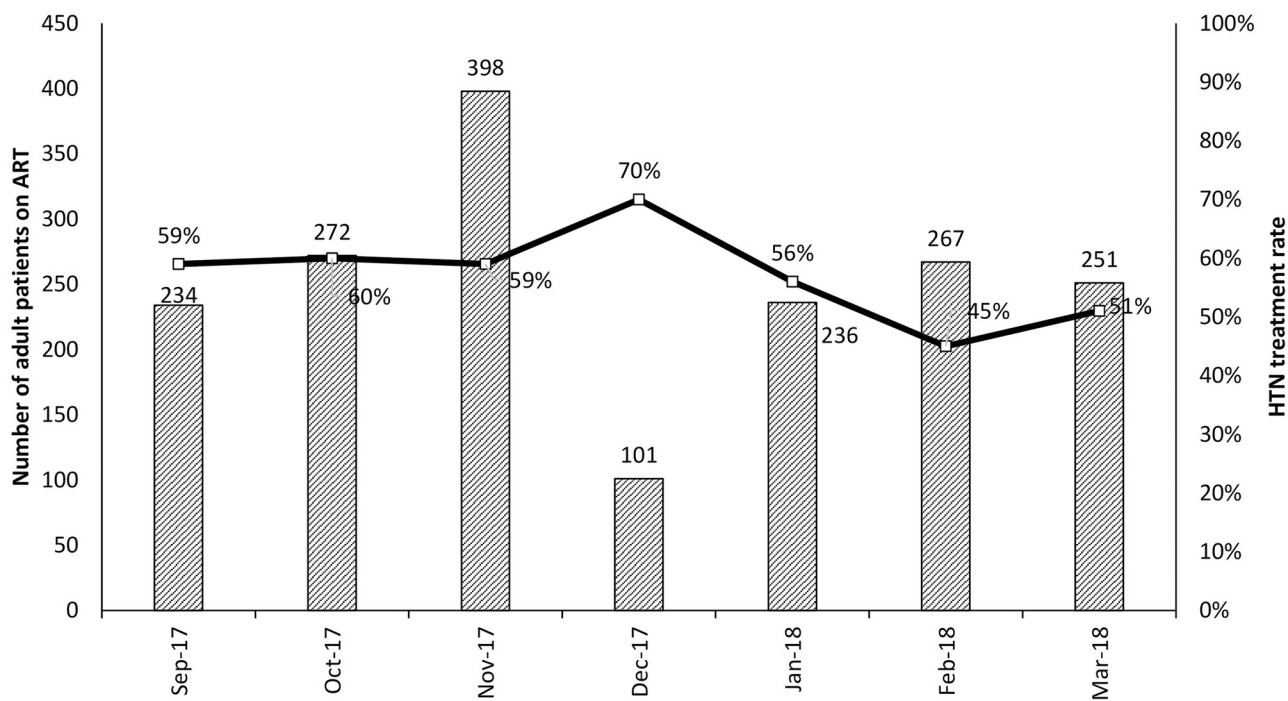

**Fig 3. HTN treatment rate and number of adult patients on ART newly diagnosed with HTN, September 2017–March 2018.** ART–antiretroviral therapy; HTN–hypertension.

clinical information systems were most common, with sites implementing, respectively, four and three interventions in each of these domains.

## Discussion

In this work, we have reported the implementation of a QIC to improve uptake of HTN screening and treatment as part of HIV care delivery. Over 12 months of implementation, HTN screening became routine in NAMPROPA sites, with coverage rates nearing 90% of all clinical encounters. Through these efforts, over 1,700 PWH were newly diagnosed with HTN, and nearly 1,000 were initiated on treatment. These results build on findings from previous work conducted in Ghana [42] and South Africa [25] that demonstrated the utility of a QIC approach in accelerating translation of clinical guidelines into practice. To our knowledge, the current work represents the first attempt to apply QIC methodology to the integration of HIV and HTN care in SSA.

Successful integration of services is a complex endeavor that requires extensive coordination, planning, and adaptation to local care contexts and resources constraints [43]. Among the challenges of integration lies the emergence of unintended, negative consequences, including increased clinic wait times [44, 45], stigma associated with care seeking for HIV and certain NCDs [46], and compromises to the effectiveness of care for existing disease-specific services. A notable strength of NAMPROPA was its concurrent collection, reporting, and continuous improvement of both HTN- and HIV-specific quality measures—a recommended approach for mitigating the impacts of unintended consequences in the context of scale up [47]. As Namibia's national electronic health system for HIV care does not contain HTN-related fields, sites implemented several interventions to enable collection and management of HTN-specific data, including development of paper-based HTN registers and documentation

**Table 2. Implemented change ideas by Chronic Care Model domain.**

| Chronic Care Model domain and definition | Intervention |
|---|---|
| Organization of health care | • Involve clinic leadership in development and monitoring of QI interventions |
| | • Review monthly performance data in QIC indicators of HTN screening and treatment during QI team meetings on a weekly basis |
| Self-management support | • Include reminders in patient health passports to follow up for repeat BP screening if indicated |
| Community linkages | • Offer HTN screening during medication pick-ups at community-based ART delivery sites |
| Delivery system redesign | • Nurses provide training to health assistants on use of automated BP machines and protocol for recording BP in registers |
| | • Re-design clinic flow to incorporate HTN screening as part of initial registration rather than during visit with nurse or physician |
| | • Streamline clinic flow to allow patients with need for follow-up BP reading, but no additional care, to present to registration for screening rather than waiting in queue for nurse or physician |
| | • Stock anti-HTN medication in HIV clinic pharmacy to streamline medication pickups |
| Decision support | • Mentors provide refresher training to nurses on national guidelines for HTN treatment, including indication for counseling on behavior modification |
| | • Mentors provide refresher training on national guidelines for referring patients to outside facilities for HTN treatment initiation |
| Clinical information systems | • Develop a blood pressure monitoring register with columns for medical record number, SBP, DBP, and whether patient is currently on HTN treatment |
| | • Record patients' BP readings in health passports and paper-based charts |
| | • Pharmacists and pharmacy assistants track stock outs of first-line anti-HTN medications |
| | • Include HTN screening data from community-based ART delivery sites in facilities' performance data |

ART–antiretroviral therapy; BP–blood pressure; DBP–diastolic blood pressure; HTN–hypertension; SBP–systolic blood pressure; QI–quality improvement; QIC–quality improvement collaborative.

of blood pressure readings in patient health passports. These interventions, while effective in the short run of the QIC, have notable limitations compared to electronic systems. Apart from enabling QI activities [48] and improving supply chain management [49], an integrated health information system can promote continuity of care across facilities—a crucial, albeit underappreciated determinant of quality [50]. Further efforts to integrate HTN and HIV care may benefit from incorporation of HTN-related fields and pharmacy dispending data into Namibia's existing health information system.

As in other countries in SSA with shortages of healthcare workers, task sharing is essential to the success of HIV service delivery in Namibia [30]. However, in the absence of policies that enable task sharing for conditions beyond HIV, these shortages stand as a formidable hindrance to the scale up of integrated care in SSA [51]. In NAMPROPA, interventions to advance task sharing task sharing included capacity building of medical assistants to measure blood pressure and nurse-physician co-management of patients with an existing HTN diagnosis. Notably, however, while clinical guidelines in Namibia allow nurses to autonomously provide the full spectrum of HIV care, only physicians can initiate HTN treatment. Particularly in the two thirds of NAMPROPA sites without an on-site physician, PWH with a new diagnosis of HTN faced significant delays in treatment initiation, with many receiving referrals to other

facilities. Policy changes to enable nurse-led HTN treatment in SSA have been considered [51], and several pilot studies have explored the feasibility of nurse-led delivery models with encouraging results [52–55]. Should nurse-led models find broad acceptance, steady attention to quality of care remains paramount [51, 56]. A structured approach, such as a QIC, may be especially useful in these settings, not only as a way to monitor the quality of scale up, but additionally as a way to promote "communities of practice" [57] through capacity-building, mentorship, and peer learning. Like all complex interventions, however, QICs require evidence-informed adaptation to be successfully implemented in new settings [58]. As numerous factors—including leadership, staff readiness, patient volume, and provider workload—can influence implementation, attention to context is crucial, and insights from implementation science may be particularly useful as implementers seek to apply a QIC approach to integrated service delivery in other settings [59].

A key finding of NAMPROPA was the consistently low rates of linkage to HTN care, with monthly rates averaging less than 60%. Low rates of linkage to HTN treatment have been reported elsewhere [60], and underscore a need to support PWH along the full "cascade" of HTN care—from screening and diagnosis to treatment and adequate control [61]. Among system-level determinants of linkage, challenges with execution and documentation of referrals for HTN treatment were most prominent. Although sites have a standardized mechanism for placing referrals—and further streamlined this process as part of the QIC—many PWH with new HTN were nevertheless deterred by the prospect of travel to an outside facility, long wait times at outpatient departments, and stock outs of anti-HTN medications. Notably, control of HTN among those on treatment was not tracked as part of NAMPROPA, although a recent cross-sectional study from Namibia suggests an unmet need to improve engagement after linkage, finding that only 43% reported "acceptable" levels of adherence (i.e., $\geq 80\%$) [62]. Like HIV, delivery of effective HTN care requires attention to a full "cascade" of essential processes, from initial screening to long-term control [61]. With insights gained through NAMPROPA, system-level interventions such as supply chain strengthening, task shifting policy reform, differentiated care models, and community awareness campaigns, stand to further improve outcomes along the HTN treatment cascade.

## Limitations

The present work has limitations. First, as data on HTN screening and treatment were not routinely collected prior to implementation of NAMPROPA, we were not able to estimate a precise baseline of performance from which to compare sites' performance. Second, as the study did not include control facilities, we cannot assess the extent to which the Hawthorne effect may have contributed to observed improvements in participating facilities. Third, as sites were purposively selected, generalizability of selected interventions to other settings may be limited without further adaptation. Fourth, although the integration of HTN screening and treatment did not adversely affect clinical HIV outcomes, we were not able to assess the QIC's effects on patient-centered outcomes such as satisfaction, wait times, and health literacy. Attention to these outcomes is an important consideration in efforts to scale up of integrated service delivery. Future work that incorporates insights from time-motion studies [44], patient-pathway analyses [63], and assessments of patient experience [64] should be pursued to better understand the effects of integration efforts on patient experience and care-seeking behaviors.

## Conclusion

As the burden of HTN grows in SSA, new, large-scale approaches are needed to sustainably offer HTN screening and treatment to all PWH. In this work, we have reported the

implementation of a QIC to incorporate HTN screening and treatment into routine HIV care across 24-high volume facilities in Namibia. Bold policies and interventions to address system-level barriers such as brittle supply chains and limited human resources will be needed to ensure the success and sustainability of integrated service delivery in SSA. As linkage to initiation of HTN treatment remained a challenge despite focused QI activities, findings of this work suggest that the uptake of HTN screening and treatment is likely to benefit from further decentralization of integrated care.

## Supporting information

**S1 Data.**
(XLSX)

**S1 File.**
(DOCX)

## Acknowledgments

The authors thank QI teams from participating sites for their dedication to continuous improvement as part of NAMPROPA. The authors also acknowledge the support of Harold Phillips, Tracey Gantt, Katie O'Connor, Simon Agolory, Sodzi Sodzi-Tettey, and Ernest Kanyoke. Portions of data from this manuscript were previously presented as part of the 35th Conference of the International Society for Quality in Healthcare, September 23–26, 2018, Kuala Lumpur, Malaysia.

## Author Contributions

**Conceptualization:** Apollo Basenero, Julie Neidel, Daniel J. Ikeda, Bruce D. Agins.

**Data curation:** Apollo Basenero, Julie Neidel, Daniel J. Ikeda.

**Formal analysis:** Apollo Basenero, Julie Neidel, Daniel J. Ikeda, Bruce D. Agins.

**Funding acquisition:** Bruce D. Agins.

**Investigation:** Apollo Basenero, Julie Neidel, Hilaria Ashivudhi, Simbarashe Mpariwa, Jacques W. N. Kamangu, Mireille A. Mpalang Kakubu, Bruce D. Agins.

**Methodology:** Apollo Basenero, Julie Neidel, Daniel J. Ikeda, Suzanne Jed, Bruce D. Agins.

**Project administration:** Apollo Basenero, Julie Neidel, Daniel J. Ikeda, Simbarashe Mpariwa, Jacques W. N. Kamangu, Mireille A. Mpalang Kakubu, Linea Hans, Gram Mutandi, Francina Tjituka, Ndapewa Hamunime, Bruce D. Agins.

**Supervision:** Apollo Basenero, Julie Neidel, Hilaria Ashivudhi, Simbarashe Mpariwa, Jacques W. N. Kamangu, Mireille A. Mpalang Kakubu, Linea Hans, Gram Mutandi, Francina Tjituka, Ndapewa Hamunime, Bruce D. Agins.

**Validation:** Simbarashe Mpariwa, Jacques W. N. Kamangu, Mireille A. Mpalang Kakubu.

**Visualization:** Daniel J. Ikeda.

**Writing – original draft:** Apollo Basenero, Julie Neidel, Daniel J. Ikeda, Bruce D. Agins.

**Writing – review & editing:** Apollo Basenero, Julie Neidel, Daniel J. Ikeda, Hilaria Ashivudhi, Simbarashe Mpariwa, Jacques W. N. Kamangu, Mireille A. Mpalang Kakubu, Linea Hans, Gram Mutandi, Suzanne Jed, Francina Tjituka, Ndapewa Hamunime, Bruce D. Agins.

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
