## [Decision Letter · Decision Letter 0]

24 Mar 2022

PONE-D-22-03989Integrating hypertension and HIV care in Namibia: a quality improvement collaborative approachPLOS ONE

Dear Dr. Agins,

Thank you for submitting your manuscript to PLOS ONE. This investigation has merits and the findings are worthy of broader dissemination. However, we feel that it does not fully meet PLOS ONE’s publication criteria as it currently stands. Therefore, we invite you to submit a revised version of the manuscript that addresses the points raised during the review process. We also ask that you adhere to our Authors Guidelines. 

We look forward to receiving your revised manuscript.

Kind regards,

Myo Minn Oo, M.D., Ph.D.

Academic Editor

PLOS ONE

Journal Requirements:

4.  Please include your tables as part of your main manuscript and remove the individual files. Please note that supplementary tables (should remain/ be uploaded) as separate "supporting information" files.

5. Acknowledgments Section: Move New Information to the Financial Disclosure:

"Thank you for stating the following in the Acknowledgments Section of your manuscript: 

[The authors thank QI teams from participating sites for their dedication to continuous improvement as part of NAMPROPA. The authors also acknowledge the support of Harold Phillips, Tracey Gantt, Katie O’Connor, Simon Agolory, Sodzi Sodzi-Tettey, and Ernest Kanyoke. The project described in this article was funded by the U.S. Department of Health and Human Services, Health Resources and Services Administration under cooperative agreement #U1NHA08599. The views expressed in this publication are solely the opinions of the authors and do not necessarily reflect the official policies of the U.S. Department of Health and Human Services, the Health Resources and Services Administration, nor does mention of the department or agency name imply endorsement by the U.S. Government. Portions of data from this manuscript were previously presented as part of the 35th Conference of the International Society for Quality in Healthcare, September 23-26, 2018, Kuala Lumpur, Malaysia.]

 [The project described in this article was funded by the U.S. Department of Health and Human Services, Health Resources and Services Administration under cooperative agreement #U1NHA08599. The funders had no role in study design, data collection and analysis, decision to publish, or preparation of the manuscript.]

6. We note you have included a table to which you do not refer in the text of your manuscript. Please ensure that you refer to Table 2 in your text; if accepted, production will need this reference to link the reader to the Table.

7. Please include a copy of Table 3 which you refer to in your text on page 8.

Additional Editor Comments:

Specifically mention your study design that is not currently clear whether it is a mixed-methods or a purely qualitative study.Add a discussion on how Hawthorne effect could impact the study's findings and what the authors did (could do) to minimise this bias.Add a "Author contributions" section before the references.Submit a STROBE report as well since this paper reports an observational study.

Reviewers' comments:

Reviewer's Responses to Questions

**Comments to the Author**

1. Is the manuscript technically sound, and do the data support the conclusions?

Reviewer #1: Partly

Reviewer #2: Yes

2. Has the statistical analysis been performed appropriately and rigorously? 

Reviewer #1: No

Reviewer #2: No

3. Have the authors made all data underlying the findings in their manuscript fully available?

Reviewer #1: Yes

Reviewer #2: Yes

4. Is the manuscript presented in an intelligible fashion and written in standard English?

Reviewer #1: Yes

Reviewer #2: Yes

5. Review Comments to the Author

Reviewer #1: Thank you the authors for reporting important services, integrating hypertension and HIV care in Namibia.

Following recommendations I would propose to improve the manuscript:

Line 53: Authors write "Implementation of a QIC enabled the integration of HTN and HIV services in Namibia". Would you please describe how did authors reach the conclusion that the QIC enabled the integration of HTN and HIV services?

Line 109: Authors shared sites were purposely selected based on the site's relative patient volumes. How do authors think purposely selection is the best way to reach the objectives of this paper? Why do you select only sites with high volumes of patients? Sites which has low performance can also provide more valuable information about uptake and also barriers authors identified.

Line 109-110: Authors shared "These sites were purposively selected based on their high relative patient volumes". As the sites purposely and better functional sites, how you can the barriers you find out will be generalizable to whole Namibia. There is a recommendation was made written in lines 300-302 based on barriers authors identified. Please share how it will work for the whole SSA.

Line 302-304: Authors should also define how findings of this work suggest that the uptake of HTN screening and treatment is likely to benefit from further decentralization of integrated care.

Reviewer #2: Integrating hypertension and HIV care in Namibia: a quality improvement collaborative approach

PONE-D-22-03989

In general, it is a little difficult to understand the processes of how to provide a quality improvement collaborative approach and how to reach the intervention package. The objective is not clear about the main outcomes to present.

Abstract

Page 2, Line 41: the term using routine measurement is not clear and could not find in the main method session.

Methods

Page 7, line 175: To capture “change ideas” and site-reported implementation barriers, who were interviewed? Is there no implementation facilitators that can share as “Change ideas”?

Page 7 line 176: how many persons interviewed?

Page 7, line 180: How did all participating teams perform Plan-Do-Study-Act (PDSA) cycles? How did you get agreement for prioritizing for the intervention?

Page 7, line 181: If there is any change package that is out of domains of the Chronic Care

Model? If yes, how it handle?

Page 6, line 153: When was final decision approved to include as the intervention? Was the final decision to approve as the intervention was after peer to peer exchange or during coaching and support?

In the regular logistic supply system, HTN drugs is already supplied for HIV clinic?

Initiate anti-HTN treatment mean it is only for first visit? Does treatment rate count the follow up visit?

It would be better to explain a little bit more about Figure 1.

It would be better to explain how interview data was recorded and analyzed it throughlly?

Results

Page 7, Line 195, what is challenges documenting for?

I think intervention that has to perform should be clearly mentioned:

Table 2: In organization of health care, what are the performance data to review?

Table 2: What is paper based reminder? How it works? It may need to explain about it.

Table 2: In community linages , offer HTN screening at community based ART delivery sites. How data of HTN screening will be included in the data at HIV clinic site?

Table 2: Whom will Mentor be provided refresher training?

Health education on HTN is not included in the intervention? Why cannot it be included?

Figure 3 2nd Y axis should be HTN treatment rate.

page 8, line 209: Reference is included in the result session?

Discussion

It would be better to discuss also about strengths and limitations of using a quality improvement collaborative approach that can affect complying the intervention package?

The purpose of integrating program is not clear. Is it for initial screening and providing initial treatment or long-term support of HIV patients with HTN?

In limitation, feasibility of health care provider to provide the services and to follow the intervention should be discussed, too.

6. PLOS authors have the option to publish the peer review history of their article (what does this mean?). If published, this will include your full peer review and any attached files.

Reviewer #1: No

Reviewer #2: **Yes: **Kyaw Ko Ko Htet

---

## [Author Response · Author response to Decision Letter 0]

22 May 2022

Journal Requirements:

1. Please ensure that your manuscript meets PLOS ONE’s style requirements, including those for file naming. 

As requested, we have prepared our revised manuscript in accordance with PLOS ONE’s style requirements. 

2. Please include a complete copy of PLOS’ questionnaire on inclusivity in global research in your revised manuscript. Our policy for research in this area aims to improve transparency in the reporting of research performed outside of researchers’ own country or community. The policy applies to researchers who have travelled to a different country to conduct research, research with indigenous populations or their lands, and research on cultural artefacts. The questionnaire can also be requested at the journal’s discretion for any other submissions, even if these conditions are not met. Please find more information on the policy and a link to download a blank copy of the questionnaire here: https://journals.plos.org/plosone/s/best-practices-in-research-reporting. Please upload a completed version of your questionnaire as Supporting Information when you resubmit your manuscript. 

As requested, we have completed the PLOS questionnaire on Inclusivity in Global Research and have included it as Supporting Information in our resubmission. 

3. Please provide additional details regarding participant consent. In the ethics statement in the Methods and online submission information, please ensure that you have specified (1) whether consent was informed and (2) wat type you obtained (for instance, written or verbal, and if verbal, how it was documented and witnessed). If your study included minors, state whether you obtained consent from parents or guardians. If the need for consent was waived by the ethics committee, please include this information. 

In line 192, we have added language to specify that the need for informed consent was waived as no patient-level data were collected. 

4. Please include your tables as part of your main manuscript and remove the individual files. Please note that supplementary table should remain/be uploaded as separate “supporting information” files. 

As requested, we have included our tables as part of the main manuscript. 

5. We note that you have included a table to which you do not refer in the text of your manuscript. Please ensure that you refer to Table 2 in your text. If accepted, production will need this reference to link the reader to Table 2. 

In response to this error, we have changed the reference in line 334 to indicate Table 2 rather than Table 3. 

6. Please include a copy of Table 3 which you refer to in your text on page 8. 

“Table 2” was erroneously referred to as “Table 3” in the text. There is no Table 3 as part of this submission. 

Additional Editor Comments: 

1. Specifically mention your study design that is not currently clear whether it is a mixed-methods or a purely qualitative study. 

We thank the editor for highlighting this omission. We have added language in the “QIC approach” section, specifically lines 127-128, to specify that the QIC is a quasi-experimental study without controls. We have also added a reference that examines approaches to QIC evaluation and highlights the inherent complexity associated with evaluating QICs. 

2. Add a discussion on how the Hawthorne effect could impact the study’s findings and what the authors did (could do) to minimize this bias. 

As our study did not use control facilities—a detail we have emphasized in the “QIC approach” section—we have added mention of the Hawthorne effect as an additional limitation of the work in lines 300-302. 

3. Add a “Authors contributions” section before the references.

As requested, we have added a “Authors contributions” section before the references. 

4. Submit a STROBE report as well since this paper reports an observational study. 

As our work describes a quality improvement intervention, our strong preference would be to structure our reporting according to the Standards for Quality Improvement Reporting Excellence (SQUIRE), as reference to these standards stands to enhance searchability and scrutiny of our work by the healthcare quality improvement field. 

Review Comments to the Author:

Reviewer #1: Thank you the authors for reporting important services, integrating hypertension and HIV care in Namibia. The following recommendations I would propose to improve the manuscript: 

Line 53: The authors write “Implementation of a QIC enabled the integration of HTN and HIV services in Namibia.” Would you please describe how the authors reached the conclusion that the QIC enabled the integration of HTN and HIV services?

In line 53, we have changed the language to specify that the QIC “provided a structured approach for integrating HTN and HIV services.” As we discuss in lines 81-88, a notable shortcoming of prior integration efforts has been a lack of attention to how a structured approach, such as a QIC, may be useful in addressing a challenge (like integration) that requires changes across multiple levels of the health system. 

Line 109: The authors share that sites were purposively selected based on the sites’ relative patient volumes. How do the authors think purposive selection is the best way to reach the objectives of this paper? Why select only sites with high volumes of patients? Sites which have low performance can also provide valuable information about uptake and barriers the authors identified. 

We agree with the reviewer that barriers and enablers to uptake of interventions in health systems are highly heterogeneous—and may vary according to patient volume. As a time-limited, donor-funded initiative, however, site selection was undertaken with the explicit aim of producing the greatest impact in the shortest period of time. To acknowledge the reviewer’s criticism, we have added the issue of spread (i.e., implementation of the QIC intervention in other sites) as an area in need of further investigation in lines 272-277. 

Lines 109-110: The authors share that “These sites were purposively selected based on their high relative patient volumes.” As the sites were purposively selected and better functional sites, how might the barriers you cite be generalizable to the rest of Namibia? There is a recommendation made in lines 300-302 based on the barriers the authors identified, please share how it will apply to SSA. 

As above, we agree with the reviewer that heterogeneity of participating sites—in factors as diverse as leadership support, staff readiness, and patient volume—can impact site performance and may similarly impact generalizability. In lines 272-277, we further acknowledge this point by emphasizing the importance of adopting a rigorous implementation science approach when considering spread of the QIC intervention to settings outside Namibia. 

Lines 302-304: The authors should also define how findings of this work suggest that the uptake of HTN screening and treatment is likely to benefit from further decentralization of integrated care. 

In response to the reviewer’s suggestion, we have added language in lines 318-319 to clarify that further decentralization may be indicated based on our finding that linkage to HTN treatment remains a challenge, even despite improvements in HTN screening coverage. 

Reviewer #2: In general, it is a little difficult to understand the processes of how to provide a quality improvement collaborative approach and how to reach the intervention package. The objective is not clear about the main outcomes to present. 

To address the reviewer’s concern, in lines 180-185 we have added additional language to provide a more detailed description of how the “change package” of interventions was constructed. In addition, we have also added a more thorough description of the QIC approach—and specifically the activities that were implemented during NAMPROPA. We hope that these changes will improve clarity regarding the main objective and outcomes of the current work. 

Page 2, Line 41: The term “routine measurement” is not clear and could not be found in the main methods section. 

In line 41 we have amended the language to specify that “routine measurement” refers to “monthly measurement of performance indicators.” 

Page 6, Line 153: When was the final decision to include a change as an intervention? Was the final decision to approve an intervention made during peer-to-peer exchange or during coaching and support?

To address the reviewer’ question, we have included additional language in lines 180-185 of the Methods section to specify that the interventions were chosen as part of the QIC’s final learning session. 

Page 7, Line 175: To capture “change ideas” and site-reported implementation barriers, who was interviewed?

We have added language in lines 176-177 to specify that 138 individuals from all participating sites were interviewed, including physicians, nurses, pharmacists, data clerks, health assistants and field officers. 

Page 7, Line 176: How many persons interviewed?

As above, we have added language in lines 176-177 to specify the number of individuals who were interviewed. 

Page 7, Line 180: How did all participating teams perform Plan-Do-Study-Act (PDSA) cycles? How did you get agreement for prioritizing the interventions?

To address the reviewers’ questions, we have language in lines 180-185 to describe the process of prioritizing interventions in more detail. 

Page 7, Line 181: Are there any changes that are outside the domains of the Chronic Care Model? If so, how were they handled?

No changes were outside the domains of the Chronic Care Model.

In the regular logistic supply system, are HTN drugs already supplied for HIV clinics?

In the supply chain system, anti-hypertensive medications are stocked in only select HIV clinics. We refer the reviewer to the section “Implementation context” where we report this in detail. 

Is initiation defined as the first visit? Or does treatment rate count follow up visits?

In response to the reviewer’s question, we refer the reviewer to the section “Data collection” were we specify that treatment initiation is defined as receipt of health education for mild hypertension and prescription of first-line monotherapy for severe hypertension. 

It would be better to explain a little bit more about Figure 1.

In response to the reviewer’s suggestion, in line 129 we have included explanation of Figure 1. 

It would be better to explain how interview data were recorded and analyzed. 

To respond to the reviewer’s suggestion, in lines 177-178 we have added details regarding how interview data were recorded and analyzed. 

Page 7, Line 195, what is challenges documenting for? I think intervention that has to perform should be clearly mentioned.

In line 203, we have changed the wording from “documenting” to “recording” to make the language clearer. 

Table 2: In organization of health care, what are the performance data to review?

To response to the reviewer’s question, we have amended the language in Table 2 to specify that “the performance data” specifically refer to monthly rates of HTN screening and treatment. 

Table 2: What is paper based reminder? How it works? It may need to explain about it. 

In response to the reviewer’s question, we have revised the language in Table 2 to specify that the “paper-based reminders” refer to reminders placed in the patients’ health passport—a document that they carry on their person to each visit. 

Table 2: In community linkages, offer HTN screening at community-based ART delivery sites. How data of HTN screening be included in the data at HIV clinic site?

In Namibia, community-based ART delivery sites are part of a hub-and-spoke model in which health facilities represent “hubs” and community-based sites represent “spokes.” Each day, healthcare workers stationed at community-based sites return to the “hubs” to report data. Thus, performance in community-based sites is included in the corresponding hub’s performance. To address this detail, we have included an additional change in Table 2 under the category of “clinical information systems.” 

Table 2: Who will mentor be providing refresher training?

We have added language in Table 2 to indicate that mentors provided refresher trainings specifically to nurses. 

Health education on HTN is not included in the intervention? Why cannot it be included?

As the definition of HTN treatment includes health education for those with mild hypertension, health education was not included as an intervention. 

Figure 2 2nd Y Axis should be HTN treatment rate. 

We have amended the 2nd Y axis to correct this error. 

It would be better to discuss also about strengths and limitations of using a quality improvement collaborative approach that can affect compiling the intervention package. 

To address the reviewer’s suggestion, in lines 305-309 under the heading “Limitations” we have added further limitations to the QIC approach. 

The purpose of integrating program is not clear. Is it for initial screening and providing initial treatment or long-term support of HIV patients with HTN?

In response to the reviewer’s question, we would argue that the answer is both to promote initial screening AND to provide long-term support to those with an existing dual diagnosis. However, as the QIC did not include indicators that capture long-term support, such as engagement and treatment success, we cannot make any conclusions about the QIC’s effect on long-term support and have described this limitation as an opportunity for future work in lines 290-295.

In limitations, the feasibility of health care providers to provide the services and to follow the intervention should be discussed too. 

We agree with the reviewer that provider workload is a central concern, particularly when considering the sustainability of QI initiatives. We have added this concern as a limitation in lines 274-277.

---

## [Decision Letter · Decision Letter 1]

5 Jul 2022

PONE-D-22-03989R1Integrating hypertension and HIV care in Namibia: a quality improvement collaborative approachPLOS ONE

Dear Dr. Agins,

Thank you for submitting your manuscript to PLOS ONE. After careful consideration, we feel that it has merit but does not fully meet PLOS ONE’s publication criteria as it currently stands. Therefore, we invite you to submit a revised version of the manuscript that addresses the points raised during the review process. While the clarity and coherence of the manuscript have improved substantially, the reviewers still have concerns in the quality and merit of the manuscript for scientific publication. Please make sure to sufficiently address or revert to all of the reviewers' comments.  Please ensure that your decision is justified on PLOS ONE’s publication criteria and not, for example, on novelty or perceived impact.

We look forward to receiving your revised manuscript.

Kind regards,

Myo Minn Oo, M.D., Ph.D.

Academic Editor

PLOS ONE

Reviewers' comments:

Reviewer's Responses to Questions

**Comments to the Author**

1. If the authors have adequately addressed your comments raised in a previous round of review and you feel that this manuscript is now acceptable for publication, you may indicate that here to bypass the “Comments to the Author” section, enter your conflict of interest statement in the “Confidential to Editor” section, and submit your "Accept" recommendation.

Reviewer #1: (No Response)

Reviewer #2: All comments have been addressed

2. Is the manuscript technically sound, and do the data support the conclusions?

Reviewer #1: Partly

Reviewer #2: Yes

3. Has the statistical analysis been performed appropriately and rigorously? 

Reviewer #1: No

Reviewer #2: Yes

4. Have the authors made all data underlying the findings in their manuscript fully available?

Reviewer #1: (No Response)

Reviewer #2: Yes

5. Is the manuscript presented in an intelligible fashion and written in standard English?

Reviewer #1: Yes

Reviewer #2: Yes

6. Review Comments to the Author

Reviewer #1: Thank you for hard work to improve the manuscript.

Authors selected study sites from only sites with high volume patients. The QIC approach may not be work in the areas with low volume patients.

About this comment, authors response was “We agree with the reviewer that barriers and enablers to uptake of interventions in health systems are highly heterogeneous—and may vary according to patient volume. As a time-limited, donor-funded initiative, however, site selection was undertaken with the explicit aim of producing the greatest impact in the shortest period of time.” This limitation reduced the strength of the findings. I still believe that the finding of this study can not be generalizable to whole Namibia. Authors should remove all statements about integration HTN and HIV services in Namibia. Policymakers and reader will misinterpret this findings. Authors should mention this limitation in the limitation section as well.

Reviewer #2: PONE-D-22-03989R1: Integrating hypertension and HIV care in Namibia: a quality improvement collaborative

Approach

I agree with the previous responses. I only have ONE minor comment.

(1) In abstract, although task sharing was recommended in the conclusion, the information of onsite physician did not mention in the abstract methods or result.

7. PLOS authors have the option to publish the peer review history of their article (what does this mean?). If published, this will include your full peer review and any attached files.

Reviewer #1: No

Reviewer #2: **Yes: **KYAW KO KO HTET

---

## [Author Response · Author response to Decision Letter 1]

5 Jul 2022

Review Comments to the Author:

Reviewer #1: Thank you for the hard work to improve the manuscript. 

The authors selected study sites from only sites with high volume patients. The QIC approach may not work in areas with low volumes of patients. About this comment, the authors respond: “We agree with the reviewer that barriers and enablers to uptake of interventions in health systems are highly heterogeneous—and may vary according to patient volume. As a time-limited, donor-funded initiative, however, site selection was undertaken with the explicit aim of producing the greatest impact in the shortest period of time.” This limitation reduced the strength of the findings. I still believe that the finding of this study cannot be generalized to all of Namibia. The authors should remove all statements about integration of HTN and HIV services in Namibia. Policymakers and readers will misinterpret these findings. The authors should mention this limitation in the limitation section as well. 

We agree with the reviewer that scale up of the QIC to other sites in Namibia would require further adaptation, with explicit attention to how facilities with lower patient volumes might perform. In response to the reviewer’s comment, we have amended language in the Abstract (line 53) and Conclusion (line 322) to specify that integration of HTN and HIV services was promoted in the 24 sites participating in the QIC—and not Namibia as a whole. We have also amended language in the Limitations section (lines 308-310) to emphasize that the generalizability of the study’s results may be limited—particularly when considering patient volumes as the reviewer mentions. 

Reviewer #2: I agree with the previous responses. I only have ONE minor comment. 

In the abstract, although task sharing was recommended in the conclusion, the information on on-site physicians was not mentioned in the abstract methods or results. 

In response to the reviewer’s comment, we have added language in the methods (lines 38-39) and results (lines 46-47) sections of the Abstract to report additional information regarding on-site physicians.

---

## [Decision Letter · Decision Letter 2]

26 Jul 2022

Integrating hypertension and HIV care in Namibia: a quality improvement collaborative approach

PONE-D-22-03989R2

Dear Dr. Agins,

We’re pleased to inform you that your manuscript has been judged scientifically suitable for publication and will be formally accepted for publication once it meets all outstanding technical requirements.

Kind regards,

Thomas Tischer

Staff Editor

PLOS ONE

Additional Editor Comments (optional):

Reviewers' comments:

Reviewer's Responses to Questions

**Comments to the Author**

1. If the authors have adequately addressed your comments raised in a previous round of review and you feel that this manuscript is now acceptable for publication, you may indicate that here to bypass the “Comments to the Author” section, enter your conflict of interest statement in the “Confidential to Editor” section, and submit your "Accept" recommendation.

Reviewer #1: All comments have been addressed

Reviewer #2: All comments have been addressed

2. Is the manuscript technically sound, and do the data support the conclusions?

Reviewer #1: Yes

Reviewer #2: Yes

3. Has the statistical analysis been performed appropriately and rigorously? 

Reviewer #1: Yes

Reviewer #2: Yes

4. Have the authors made all data underlying the findings in their manuscript fully available?

Reviewer #1: (No Response)

Reviewer #2: Yes

5. Is the manuscript presented in an intelligible fashion and written in standard English?

Reviewer #1: (No Response)

Reviewer #2: Yes

6. Review Comments to the Author

Reviewer #1: (No Response)

Reviewer #2: PONE-D-22-03989R2: Integrating hypertension and HIV care in Namibia: a quality improvement collaborative approach

Thanks authors. I agree with your responses on previous comments. I do not have any comment on it.

7. PLOS authors have the option to publish the peer review history of their article (what does this mean?). If published, this will include your full peer review and any attached files.

Reviewer #1: No

Reviewer #2: **Yes: **Kyaw Ko Ko Htet

---

## [Editor Report · Acceptance letter]

3 Aug 2022

PONE-D-22-03989R2 

Integrating hypertension and HIV care in Namibia: a quality improvement collaborative approach 

Dear Dr. Agins:

I'm pleased to inform you that your manuscript has been deemed suitable for publication in PLOS ONE. Congratulations! Your manuscript is now with our production department. 

Kind regards, 

on behalf of

Dr. Thomas Tischer 

Staff Editor

PLOS ONE